 

# NOTCH activity differentially affects alternative cell fate acquisition and maintenance

**Leonard Cheung[1], Paul Le Tissier[2], Sam GJ Goldsmith[3], Mathias Treier[4,5], Robin Lovell-Badge[3], Karine Rizzoti[3]***

[1]Department of Human Genetics, University of Michigan, Ann Arbor, United States; [2]Centre for Discovery Brain Science, Integrative Physiology, Edinburgh, United Kingdom; [3]The Francis Crick Institute, London, United Kingdom; [4]Cardiovascular and Metabolic Sciences, Max Delbrück Center for Molecular Medicine in the Helmholtz Association (MDC), Berlin, Germany; [5]Charité-Universitätsmedizin Berlin, Berlin, Germany

**Abstract** The pituitary is an essential endocrine gland regulating multiple processes. Regeneration of endocrine cells is of therapeutic interest and recent studies are promising, but mechanisms of endocrine cell fate acquisition need to be better characterised. The NOTCH pathway is important during pituitary development. Here, we further characterise its role in the murine pituitary, revealing differential sensitivity within and between lineages. In progenitors, NOTCH activation blocks cell fate acquisition, with time-dependant modulation. In differentiating cells, response to activation is blunted in the POU1F1 lineage, with apparently normal cell fate specification, while POMC cells remain sensitive. Absence of apparent defects in *Pou1f1-Cre; Rbpj^{fl/fl}* mice further suggests no direct role for NOTCH signalling in POU1F1 cell fate acquisition. In contrast, in the POMC lineage, NICD expression induces a regression towards a progenitor-like state, suggesting that the NOTCH pathway specifically blocks POMC cell differentiation. These results have implications for pituitary development, plasticity and regeneration. Activation of NOTCH signalling in different cell lineages of the embryonic murine pituitary uncovers an unexpected differential sensitivity, and this consequently reveals new aspects of endocrine lineages development and plasticity.
DOI: https://doi.org/10.7554/eLife.33318.001

*For correspondence:
Karine.Rizzoti@crick.ac.uk

**Competing interests:** The authors declare that no competing interests exist.

## Introduction

The pituitary is an essential endocrine gland that is closely associated with the hypothalamus, in the ventral diencephalon, which controls its hormonal secretions. Pituitary hormones regulate different aspects of metabolism hence they are involved in maintenance of organism homeostasis, and important physiological functions such as growth, reproduction and stress. In consequence, pituitary hormone deficiencies, either congenital (1 in 3500 to 10000 births) (*Alatzoglou and Dattani, 2009*) or acquired later in life, mostly caused by tumours or brain damage, are associated with significant morbidity. We, and others, have characterized a population of stem cells (SCs) in the mouse pituitary (*Rizzoti et al., 2013*) (*Andoniadou et al., 2013*). In addition, the differentiation of embryonic SC to generate endocrine cells has been demonstrated in vitro (*Suga et al., 2011*). It would be of therapeutic interest to utilize SCs to generate endocrine cells; however, the limited efficiency of current protocols highlights the requirement for improved characterization of the mechanisms of endocrine cell fate acquisition.

The pituitary rudiment, or Rathke's Pouch (RP), develops from the oral ectoderm. It comprises progenitors that will give rise to the pituitary anterior lobe (AL) ventrally, and the intermediate lobe (IL) dorsally. The AL comprises five pituitary endocrine cell types (somatotrophs, lactotrophs, thyrotrophs, corticotrophs and gonadotrophs), while the IL contains exclusively melanotrophs (for review see (*Rizzoti, 2015*)). The third lobe, or posterior lobe (PL), is of neural origin and is comprised of glial cells and vasopressin and oxytocin hypothalamic axon terminals. During early pituitary development, RP progenitors initially proliferate, then gradually commit towards endocrine cell fates. Cell fate acquisition is associated with exit from mitosis (*Bilodeau et al., 2009*) and an Epithelial to Mesenchymal Transition (EMT)-like process (*Pérez Millán et al., 2016*). Birth-dating studies have moreover shown that the different endocrine lineages emerge during the same period, mostly between 11.5 and 13.5dpc in the mouse (*Davis et al., 2011*). The molecular events underlying endocrine differentiation have been relatively well described for each endocrine cell type. Within each lineage, expression of specific transcription factors is initiated as progenitors commit to an endocrine fate, such as TBX19 (also known as TPIT) along with NEUROD1 in future corticotrophs or with PAX7 in prospective melanotrophs, POU1F1 (also known as PIT1) in future somatotrophs, lactotrophs and thyrotrophs, and SF1 for gonadotroph cell fate (for review see (*Rizzoti, 2015*)). It is still unclear however how the endocrine cell fates are initially specified, but the NOTCH pathway is known to be involved.

The NOTCH pathway plays a central role during cell fate acquisition in different organs. In vertebrates, the ligands, Delta-like or Jagged proteins, activate the transmembrane receptors NOTCH1-4. Following activation, cleavage of the NOTCH Intracellular Domain (NICD) allows its translocation to the nucleus, resulting in activation of target genes, such as the HES bHLH transcription factors, by the pathway nuclear effector RBPJ. Deregulation of the pathway is associated with inherited and degenerative diseases, but also with cancers in humans (for review see (*Hori et al., 2013*)). In the pituitary, the receptors NOTCH 2 and 3, the ligands Jagged one and Delta-like 1, and downstream effectors of the HES and HEY families are expressed in embryonic progenitors; as endocrine lineages emerge expression of these genes is mostly confined to SCs and this persists post-natally (*Chen et al., 2006*; *Raetzman et al., 2004*; *Zhu et al., 2006*). In contrast, the ligands Dll1 and 3, and HES6, that does not bind DNA itself but suppresses HES1 activity, are present in differentiating cells, suggesting that the pathway is also active once cell fate is acquired (*Raetzman et al., 2004*). Studies investigating the consequences of conditional deletion of RBPJ (*Zhu et al., 2015*; *Zhu et al., 2006*), *Notch2* and loss of the NOTCH targets *Hes1* and *Hes5* (*Kita et al., 2007*; *Raetzman et al., 2007*) (*Nantie et al., 2014*) support a role for NOTCH pathway in maintenance of an undifferentiated proliferative state to allow emergence of the different endocrine cell types. In contrast, overactivation of the pathway by conditional expression of NICD in either committed progenitors (*Zhu et al., 2006*), or differentiated corticotrophs and melanotrophs constituting the POMC lineage (*Goldberg et al., 2011*), results in a blockade of cell differentiation.

To better characterize the role of the NOTCH pathway during pituitary development, we have here manipulated its activity and compared outcomes in different cellular contexts. Using *Sox2-CreERT2* (*Arnold et al., 2011*) and *Nkx3.1Cre* (*Lin et al., 2007*), we show that progenitors are particularly sensitive to NOTCH signalling, as cell fate acquisition is mostly prevented by NOTCH overactivation. However, we reveal that timing and/or duration of activation modulates cell responses; early activation results in exclusion of cells from the future IL, while activated cells remain in the IL if induction is performed 72 hr later. In contrast, in POU1F1 positive committed cells, NICD expression results in a blunted activation of NOTCH target genes. In consequence, there is no apparent effect on differentiation of somatotrophs, thyrotrophs and lactotrophs. However post-natally, as activation becomes more efficient, there is a reduction in Growth Hormone (GH) pituitary contents, suggesting that the function of GH-secreting somatotrophs is altered. Nonetheless, and in agreement with a minor role of NOTCH pathway in this lineage, we observe that deletion of *Rbpj* using the same POU1F1-Cre does not affect GH levels. Intrigued by the relatively modest effect of NOTCH activation in the POU1F1 lineage, we expressed NICD in the POMC lineage, where we observe an efficient activation of the pathway, showing that corticotrophs and melanotrophs remain sensitive to NOTCH activation. While cell fate acquisition did not appear affected initially, we observe a fast downregulation of differentiation markers expression, while SC markers are up-regulated, as well as a spectacular regression of IL soon after birth. This study uncovers an unexpected differential sensitivity to NOTCH activity according to timing and lineage identity. We propose that the sensitivity of the

POMC lineage to NICD activity reflects a specific physiological requirement of NOTCH pathway to prevent differentiation toward the first endocrine cell lineage to emerge, the corticotrophs. Moreover, the lasting sensitivity of this lineage may have a pathological relevance because NOTCH activation has been associated with tumorigenesis.

## Results

### Activation of NOTCH pathway in RP progenitors efficiently blocks cell fate acquisition

We initially examined cell fate acquisition potential after over-activation of the NOTCH pathway in RP progenitors. For this purpose, we used $Sox2^{CreERT2}$ (*Arnold et al., 2011*) and $Nkx3.1^{Cre}$ (*Lin et al., 2007*) to induce recombination of $Rosa26^{floxSTOP-NICD1}$ (*Murtaugh et al., 2003*) and $Rosa26^{floxSTOP-NICD2}$ (*Fujimura et al., 2010*) alleles. We used $Rosa26^{floxSTOP-NICD1}$ because other studies where NOTCH pathway was activated utilised this allele (*Goldberg et al., 2011*; *Zhu et al., 2006*). However, NOTCH1 is not expressed in the pituitary while NOTCH2 is, so we also used $Rosa26^{floxSTOP-NICD2}$ (*Raetzman et al., 2004*). SOX2 is present in all pituitary progenitors (*Fauquier et al., 2008*; *Rizzoti et al., 2013*), while NKX3.1 is expressed predominantly in those located in the dorsal pituitary, peaking at 12.5dpc (*Treier et al., 1998*) (*Goldsmith et al., 2016*). Lineage tracing analysis of NKX3.1Cre confirms that this driver is predominantly active in future melanotrophs, and also, albeit less efficiently, in all anterior lobe endocrine populations (*Figure 1A–B*).

We first verified that the NOTCH pathway was activated following induction of NICD expression in $Nkx3.1^{Cre/+};Rosa26^{floxSTOP-NICD1/+}$ and $Nkx3.1^{Cre/+};Rosa26^{floxSTOP-NICD2/+}$ pituitaries. We quantified expression levels for several NOTCH target gene transcripts at 18.5dpc (*Figure 1C*). We used quantitative PCR (RT-qPCR) and observed a significant induction of targets of the *Hes* and *Hey* families, showing that NOTCH pathway can be efficiently over-activated in RP progenitors. We observe a particularly potent induction of *Hes5*. *Hes1* and *Hes5* are closely related and are frequently expressed in complementary patterns, in particular in the developing nervous system (*Hatakeyama et al., 2004*). In the developing pituitary, *Hes5* expression has previously been described as undetectable, but it is up-regulated upon deletion of *Hes1* and compensates for its loss (*Kita et al., 2007*). Therefore induction of *Hes5* expression in $Nkx3.1^{Cre/+};Rosa26^{floxSTOP-\ NICD1\ or\ 2/+}$ is physiologically relevant.

We then analysed expression levels of markers for endocrine and progenitor cells in $Nkx3.1^{Cre/+};$ $Rosa26^{floxSTOP-\ NICD1\ or\ 2/+}$ pituitaries at E18.5 by RT-qPCR (*Figure 1C–D*). The transcription factor PROP1 is expressed in pituitary progenitors where it is required for cell fate progression toward commitment (*Pérez Millán et al., 2016*) and has been proposed to be a direct target of NOTCH pathway (*Zhu et al., 2006*). In agreement with these data, we observe an upregulation of its expression upon NOTCH activation, which only reaches significance using the NICD2 allele (*Figure 1C*). *Prop1* is rapidly down-regulated as cells differentiate (*Yoshida et al., 2009*; *Yoshida et al., 2011*), therefore, up-regulation of its expression suggests an impairment in cell differentiation. Persistent NICD1, but not NICD2, expression leads indeed to a significant reduction in *Pou1f1*, which encodes a transcription factor required for maintenance of GH (growth hormone), PRL (prolactin secreted by lactotrophs) and TSH (thyroid-stimulating hormone secreted by thyrotrophs) secreting cells (*Bodner et al., 1988*) (*Figure 1D*). In addition, we also observe a reduction in the expression of *Pax7*, the product of which is a pioneer factor for melanotroph fate (*Budry et al., 2012*), and *Pomc* (proopiomelanocortin), which encodes the precursor of the hormone secreted by corticotrophs (ACTH, adrenocorticotropic hormone) and melanotrophs (MSH, melanocyte stimulating hormone) when either NICD1 or NICD2 are overexpressed (*Figure 1D*). In contrast, expression of *Sox2* is not significantly altered. In summary, these results indicate that melanotrophs, endocrine cells of the POU1F1 lineage, and possibly corticotrophs, are affected by NOTCH over-activation. In addition, NICD1 appears more efficient, and/or has a specific effect on the POU1F1 lineage that we do not observe using NICD2 (*Figure 1D*), suggesting non-redundant function for NOTCH cellular domains.

We subsequently performed 3D reconstitution of $Nkx3.1^{Cre/+};Rosa26^{floxSTOP\ NICD1\ or\ 2/+}$ pituitaries at 15.5dpc (*Figure 1E*). We observe clear malformations of the gland with varying degrees of severity. Using NICD1, 2 out of 7 mutant embryos presented a severe phenotype, while using NICD2, 3 out of 5 were severely affected, suggesting that NICD2 was associated with more serious defects.

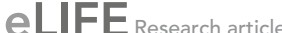

**Figure 1.** Activation of NOTCH pathway in progenitors efficiently blocks cell fate acquisition. (**A**) Immunofluorescence of 18.5dpc *Nkx3.1^{Cre/+}*; *Rosa26^{ReYFP/+}* pituitaries. This lineage tracing analysis shows that *NKX3.1^{Cre}* is active in precursors for different endocrine cells. (**B**) Percentage of eYFP positive endocrine cells for each hormone-secreting population in 18.5dpc *Nkx3.1^{Cre/+}; Rosa26^{ReYFP/+}* 18.5 dpc pituitaries. *Nkx3.1^{Cre}* is mostly active in IL precursors with POMC;eYFP double positive representing 92% (±1.8 SD, n = 3) of POMC positive melanotrophs, and 55.4% (±3.6 SD, n = 3) of POMC positive corticotrophs. The other AL cell types are also present in the progeny of NKX3.1 expressing progenitors with GH;eYFP double positive

*Figure 1 continued on next page*

Figure 1 continued

representing 51.9% (±2 SD, n = 2) of somatotrophs, PRL;eYFP double positive representing 34.6% (±2.4 SD, n = 3) of lactotrophs, TSH;eYFP double positive representing 31.7% (±4.2 SD, n = 3) of thyrotrophs and LH;eYFP double positive representing 38.2% (±7.1 SD, n = 3) of gonadotrophs. (C) RT-qPCR analysis of NOTCH target genes in $Nkx3.1^{Cre/+}$;$Rosa26^{flSTOPNICD1/+}$ and $Nkx3.1^{Cre/+}$;$Rosa26^{flSTOPNICD2/+}$ pituitaries at 18.5dpc (n = 3 to 4 embryos/genotype, $Nkx3.1^{Cre/+}$;$Rosa26^{flSTOPNIC/+D}$ –T for NICD transgenic - versus $Rosa26^{flSTOPNICD/+}$ -NT for non-NICD transgenic- littermates, unpaired t test performed). There is a significant induction of NOTCH pathway target gene expression, particularly of $Hes5$. (D) RT-qPCR analysis of pituitary cell-type markers in $Nkx3.1^{Cre/+}$;$Rosa26^{flSTOPNICD1/+}$ and $Nkx3.1^{Cre/+}$;$Rosa26^{flSTOPNICD2/+}$ pituitaries at 18.5dpc (n = 3 to 6 embryos/genotype, unpaired t test performed). POMC lineage markers ($Pax7$ and $POMC$) are affected by NICD1 and 2 over-expression, while $Pit1$ and $Gh$ are only affected by NICD1 induction. (E) 3D reconstruction of 15.5dpc $Nkx3.1^{Cre/+}$;$Rosa26^{flSTOPNICD/+}$ pituitaries. Soft tissues are false-coloured in green and the basisphenoid bone in red. Severity of the phenotype is variable with most affected embryos showing misfolding of the epithelium lining the Rathke's pouch lumen, reduction of the lateral pituitary wings and no midline fusion of the basisphenoid bone. (F) Immunofluorescence of 18.5dpc $Sox2^{CreERT2/+}$; $Rosa26^{ReYFP}$ and $Sox2^{CreERT2/+}$;$Rosa26^{flSTOPNICD1/+}$ pituitaries induced at 9.5dpc. There is a mosaic pattern of eYFP/GFP expression with eYFP expression being more robust than GFP. In contrast with eYFP in control embryos, most NICD1iresGFP positive cells appear SOX2 positive. In addition, GFP positive cells do not express GH or POMC. Finally, there are no GFP positive cells in IL. (G) Immunofluorescence of 18.5dpc $Nkx3.1^{Cre/+}$; $Rosa26^{ReYFP}$ and $Nkx3.1^{Cre/+}$;$Rosa26^{flSTOPNICD1/+}$. IL in $Nkx3.1^{Cre/+}$;$Rosa26^{flSTOPNICD1/+}$ is misfolded and hypomorphic. GFP positive cells, both in IL and AL appear to retain high levels of SOX2. Scale bars represent 100 µm for A and 50 µm for F and G. IL is underlined. H.Quantification of the proportion of recombined cells maintaining expression of SOX2 in AL at 18.5dpc. There is a significant increase in the proportion of recombined cells retaining SOX2 expression when NICD is present (n = 3 pituitary/genotype). Moreover this effect is significantly more robust in $Sox2^{CreERT2/+}$;$Rosa26^{flSTOPNICD1/+}$ compared to $Nkx3.1^{Cre/+}$;$Rosa26^{flSTOPNICD1/+}$. Data are presented as mean ± SD, unpaired t test performed, p=0.0003 when comparing $Sox2^{CreERT2/+}$; $Rosa26^{ReYFP/+}$ and $Sox2^{CreERT2/+}$;$Rosa26^{flSTOPNICD1/+}$, p=0.0001 when comparing $Nkx3.1^{Cre/+}$;$Rosa26^{ReYFP}$ and $Nkx3.1^{Cre/+}$;$Rosa26^{flSTOPNICD1/+}$ and 0.0299 when comparing $Sox2^{CreERT2/+}$;$Rosa26^{flSTOPNICD1/+}$ and $Nkx3.1^{Cre/+}$;$Rosa26^{flSTOPNICD1/}$. P is calculated after angular transformation of percentages, n = 3 for each genotype.

DOI: https://doi.org/10.7554/eLife.33318.002

The following source data and figure supplements are available for figure 1:

**Source data 1.** Countings for graph H.
DOI: https://doi.org/10.7554/eLife.33318.005

**Figure supplement 1.** IL recombined cells in 12.5dpc $Sox2^{CreERT2/+}$;$Rosa26^{flSTOPNICD1/+}$ Rathke's pouch are not eliminated by apoptosis.
DOI: https://doi.org/10.7554/eLife.33318.003

**Figure supplement 2.** SOX9 is up-regulated in $Sox2^{CreERT2/+}$;$Rosa26^{flSTOPNICD1/+}$ progenitors.
DOI: https://doi.org/10.7554/eLife.33318.004

However, this effect may be confounded by varying levels of Cre activity because other studies utilizing Nkx3.1$^{Cre}$ have also reported variations in phenotype severity such as prostate defects in $Nkx3.1^{Cre}$;$Fgfr2^{flox/flox}$ mutants (**Lin et al., 2007**), and spinal defects in $Nkx3.1^{Cre}$;$Erk5^{flox/flox}$ animals (**Loveridge et al., 2017**). The AL appears smaller, particularly the lateral wings of the developing gland. In addition, the progenitor zone, surrounding the pituitary cleft appears enlarged and folded, suggesting a retention of progenitors. The sphenoid bone is sometimes affected (**Figure 1E**, high severity phenotype) and this could be a direct consequence of the activity of NKX3.1Cre in the developing skeleton, or indirectly because of the abnormal development of the gland, as we observed previously in $Sox2^{fl/fl}$;$Nkx3.1^{Cre/+}$ embryos (**Goldsmith et al., 2016**). Over-activation of NOTCH signalling in the developing skeleton causes severe developmental abnormalities (**Hosaka et al., 2013**) and this likely explains the lethality of $Nkx3.1^{Cre/+}$;$Rosa26^{floxSTOPNICD1 \ or \ 2/+}$ newborns.

We next assessed cell fate acquisition potential in mutant pituitaries by analysing the number of NICD1 expressing cells maintaining expression of SOX2 at 18.5dpc (**Figure 1F–H**). We restricted here our analysis to NICD1 as this allele seemed to affect endocrine lineages more efficiently than NICD2 (**Figure 1D**). We can identify cells in which $Rosa26^{floxSTOPNICD1}$ has been recombined because they express GFP, as the NICD allele includes an IRES-nuclearGFP cassette (**Murtaugh et al., 2003**) (**Figure 1F**). We analysed embryos expressing $Sox2^{CreERT2}$, induced at 9.5dpc, or $Nkx3.1^{Cre}$ in combination with $Rosa26^{floxSTOPNICD1}$ and, as controls, with the Cre reporter allele $Rosa26^{ReYFP}$(**Srinivas et al., 2001**)). Of note, cellular patterns of NICD1iresGFP and eYFP expression differ because GFP is nuclear while eYFP is present throughout the cell, resulting in apparently increased levels of eYFP. However, we observed that the number of fluorescent cells resulting from recombination of the $Rosa26^{ReYFP}$ allele always appears higher than that of GFP from $Rosa26^{floxSTOPNICD1}$, particularly when we use an inducible Cre (**Figure 1F**). This is therefore likely due to a difference in recombination efficiency between the two alleles. In addition, GFP in $Rosa26^{floxSTOPNICD1}$ is

preceded by an IRES and this could result in reduced expression levels. We did not observe any TUNEL activity so this difference is not due to a loss of GFP positive cells by apoptosis (*Figure 1—figure supplement 1*). In control Cre; *Rosa26$^{ReYFP/+}$* embryos quantification revealed only a small proportion, less than 10%, of recombined cells retaining expression of SOX2 in AL (we excluded the cleft region where SOX2 positive progenitors are maintained), because the vast majority of early progenitors have differentiated at this stage (*Rizzoti et al., 2013*). In contrast, in either *Nkx3.1$^{Cre/+}$* or *Sox2$^{CreERT2/+}$;Rosa26$^{floxSTOPNICD1/+}$* embryos, the majority of recombined cells maintain SOX2 expression and do not express endocrine differentiation markers such as GH and POMC (*Figure 1*. F-H and *Figure 2A*). Expression of SOX9, another marker of pituitary SC (*Fauquier et al., 2008*; *Rizzoti et al., 2013*), is observed in most NICD1iresnucGFP positive cells (*Figure 1—figure supplement 2*). Therefore, over-activation of the NOTCH pathway in RP progenitors efficiently blocks cell fate acquisition. In addition, timing and/or duration of NOTCH activation appear important, because the earliest it is induced, comparing *Sox2$^{CreERT2}$* induced at 9.5dpc (84% SOX2;GFP double positive/ GFP at 18.5dpc) and *Nkx3.1$^{Cre}$*, active at 12.5dpc (60% SOX2;GFP double positive/GFP at 18.5dpc), the more potent is its effect (*Figure 1H*).

## Temporal modulation of over-activation of NOTCH in progenitors has contrasting phenotypic consequences on IL development

When comparing *Sox2$^{CreERT2}$* and *Nkx3.1Cre;Rosa26$^{floxSTOPNICD1}$* pituitaries at 18.5dpc we noticed an obvious difference in the IL (*Figure 1F,G*, lower panels). NKX3.1Cre is mostly active in dorsal RP and cell recombination is essentially ubiquitous in the prospective IL. As a consequence, and in agreement with a blockade of cell differentiation induced by NOTCH over-activation, melanotrophs are dramatically reduced in *Nkx3.1Cre;Rosa26$^{floxSTOP-NICD1}$* (*Figure 2A*) while instead SOX2 is maintained at high levels (*Figure 1G*). However, some cells are still able to commit to a melanotroph fate, because a proportion of NICD1iresnucGFP positive cells express low levels of PAX7, but are unable to upregulate POMC (*Figure 2B*). In sharp contrast, we do not observe any NICD1iresnucGFP-positive cells in *Sox2$^{CreERT2}$;Rosa26$^{floxSTOPNICD1}$* IL, when Cre is induced at 9.5dpc (*Figure 1F* lower panel). This is clearly due to NOTCH activation, because in *Sox2$^{CreERT2/+}$;Rosa26$^{ReYFP/+}$* pituitaries, IL comprises eYFP-positive cells, which as expected reveal a mosaic pattern of induction of cell recombination following tamoxifen induction (*Figure 1F* upper panel).

The differential distribution of NICD1iresnucGFP-positive cells in *Sox2$^{CreERT2}$* and *Nkx3.1$^{Cre}$; Rosa26$^{floxSTOPNICD1}$* IL could be due to a difference in the pattern of recombination. This is because *Nkx3.1$^{Cre}$* is active homogenously across what will become the IL while *Sox2$^{CreERT2}$* is induced in a mosaic pattern. Alternatively, or in addition, the difference could be explained by the variation in timing of activation because *Nkx3.1$^{Cre}$* is mostly active at 12.5dpc, 72 hr after we induced *Sox2$^{CreERT2}$* at 9.5dpc. To discriminate between these two explanations, we examined *Sox2$^{CreERT2}$; Rosa26$^{floxSTOPNICD1}$* IL following induction at 12.5dpc, mimicking *Nkx3.1$^{Cre}$* timing, but in a mosaic pattern. At 18.5dpc we observe NICD1iresnucGFP positive cells in IL (*Figure 2C*). In addition, similarly to what we saw in *Nkx3.1$^{Cre}$;Rosa26$^{floxSTOP-NICD1}$* IL, these cells are unable to upregulate POMC expression, but some commit to a melanotroph fate and express low levels of PAX7 (*Figure 2D*). Therefore, it is the variation in timing of activation in dorsal RP that underlines the phenotypic differences observed between *Sox2$^{CreERT2}$;Rosa26$^{floxSTOPNICD1}$* induced at 9.5dpc and *Nkx3.1$^{Cre}$; Rosa26$^{floxSTOPNICD1}$*.

To understand how recombined cells disappear from the IL of *Sox2$^{CreERT2/+}$;Rosa26$^{floxSTOPNICD1}$* induced at 9.5dpc, we harvested pituitaries earlier during embryogenesis (*Figure 2E*). We never detected TUNEL-positive cells in IL (*Figure 1—figure supplement 1*). However, at 12.5dpc we observed IL NICD1iresnucGFP-positive cells and these are essentially restricted to the ventral most part of the developing IL, while control embryos show an equal distribution across IL (*Figure 2G*). More precisely, we observe patches of IL NICD1iresnucGFP positive cells apparently reaching toward the AL progenitor layer (*Figure 2E*). We do also observe regions of apparent contact between the future IL and AL in control embryos, but these are rare (*Figure 2—figure supplement 1*). To assess epithelial integrity in mutant cells, we examined E-cadherin and ZO-1 expression, present respectively in adherent and tight junctions. Alterations in localisation of both proteins, normally present in the apical lateral membranes, are observed with a relocalisation around the plasma membrane, and sometimes loss from the apical ones, in IL NICD1iresnucGFP-positive cells closest to Rathke's cleft. This suggests a loss of epithelial polarity of mutant cells (*Figure 2E,F*). Experimental

**Figure 2.** Differential timing of NOTCH activation IL progenitors has contrasting phenotypic consequences. (**A**) Immunofluorescence of 18.5dpc *Nkx3.1^{Cre/+};Rosa26^{ReYFP}* and *Nkx3.1^{Cre/+};Rosa26^{flSTOPNICD1/+}* pituitaries. NICD1iresGFP positive cells do not express POMC in contrast with eYFP positive cells in control embryos. (**B**) Immunofluorescence of 18.5dpc *Nkx3.1^{Cre/+};Rosa26^{ReYFP}* and *Nkx3.1^{Cre/+};Rosa26^{flSTOPNICD1/+}*. In control embryos, IL melanotrophs express PAX7 and POMC while NICD1iresGFP positive cells in mutants are unable to upregulate POMC. Some show low levels of PAX7 expression (arrowhead) suggesting commitment toward melanotroph fate. A few cells upregulate PAX7 and POMC in mutants, but are NICD1iresGFP negative non-recombined cells (arrows). (**C, D**) Immunofluorescence of 18.5dpc *Sox2^{CreERT2/+};Rosa26^{flSTOPNICD1/+}*pituitaries induced at 12.5dpc. NICD1iresGFP positive cells are now observed in IL. These express SOX2, but are POMC negative (high magnification inset). Similarly to what is observed in *Nkx3.1^{Cre/+};Rosa26^{flSTOPNICD1/+}*embryos, some NICD1iresGFP cells express PAX7 (D, arrowhead). (**E**) Immunofluorescence of 12.5dpc *Sox2^{CreERT2/+};Rosa26^{flSTOPNICD1/+}* induced at 9.5dpc. Lower panel represents a magnification of the boxed area. NICD1iresGFP IL positive cells appear clustered ventrally toward AL. E-cadherin staining is lost from the apical membranes of NICD;GFP positive cells. F-actin is stained using phalloidin. (**F**) Immunofluorescence of 12.5dpc *Sox2^{CreERT2/+};Rosa26^{flSTOPNICD1/+}* induced at 9.5dpc. Lower panel represents a magnification of the boxed area. Expression of ZO-1, along with that of E-cadherin is lost/altered from the apical membranes of the NICD;GFP positive cells that are closest to the cleft. Therefore, both tight and adherent junctions appear abnormal in these cells, suggesting a loss of epithelial polarity. (**G**) Proportion of recombined cells

*Figure 2 continued on next page*

*Figure 2 continued*

located in the ventral half of IL in 12.5dpc *Sox2^{CreERT2/+};Rosa26^{ReYFP}* and *Sox2^{CreERT2/+};Rosa26^{flSTOPNICD1/+}* pituitaries induced at 9.5dpc. In control *Sox2^{CreERT2/+};Rosa26^{ReYFP}* pituitaries, eYFP positive cells are homogeneously distributed in IL (n = 5 embryos, 58 (±8.5 SD) cells were counted/embryo). In contrast, 85% of NICD1iresGFP positive cells in mutant embryos are found in the ventral half of IL (n = 5 embryos, 29 (±23.6 SD) cells were counted/embryo). Unpaired t test performed, p=0.0045 after angular transformation of percentages. An annotated picture illustrates how cells were counted in dorsal and ventral IL. Scale bars represent 50 μm for A, C and D (upper panel), 20 μm for B and 10 μm for E (magnification) and F. IL is underlined.
DOI: https://doi.org/10.7554/eLife.33318.006
The following source data and figure supplement are available for figure 2:

**Source data 1.** Countings for graph G.
DOI: https://doi.org/10.7554/eLife.33318.008
**Figure supplement 1.** Contact between prospective IL and AL in control embryos.
DOI: https://doi.org/10.7554/eLife.33318.007

induction of cell fate change within epithelial cells is known to result in cell extrusion (*Bell and Thompson, 2014*; *Bielmeier et al., 2016*). We suggest that in IL, over-activation of NOTCH pathway, resulting in blockade of cell commitment and therefore singling out mutant cells, induces an apical extrusion of recombined cells toward AL. This phenomenon only happens in a narrow time window because overactivation performed 72 hr later still leads to a blockade of cell fate acquisition, but cell extrusion is no longer observed, or is significantly less efficient. Therefore, variations in timing and/or duration of NOTCH pathway activation result in strikingly different outcomes during IL development, reflecting modulation of cell responses to NOTCH activation.

## Expression of NICD in the POU1F1 lineage does not alter cell fate acquisition, but post-natal function of somatotrophs is altered

We then decided to examine the effects of NOTCH overactivation in committed cells. POU1F1 is a homeodomain transcription factor, exclusively expressed in the pituitary where it is required for the post-natal expansion of somatotrophs, lactotrophs and thyrotrophs (*Ward et al., 2006*). It starts to be expressed at 13.5dpc, marking cell commitment, and its expression is maintained in differentiated somatotrophs, lactotrophs and thyrotrophs.

We thus generated *Pou1f1-Cre;Rosa26^{flSTOPNICD1/+}* and *Pou1f1-Cre;Rosa26^{flSTOPNICD2/+}* animals. These are born and they display essentially normal growth curves, except for a transient delay observed exclusively in males, which only reached significance in *Pou1f1-Cre; Rosa26^{flSTOPNID2/+}* animals (*Figure 3A*). We verified by RT-QPCR activation of NOTCH target genes and observed a potent up-regulation of *Hes5* in adult *Pou1f1-Cre; Rosa26^{flSTOPNICD1and 2/+}* pituitaries demonstrating that the pathway is activated (*Figure 3B*).

Hormonal levels could be affected without altering animal weight therefore, we examined GH pituitary contents by sandwich ELISA. We detected a significant decrease of pituitary GH in both *Pou1f1-Cre;Rosa26^{flSTOPNICD1/+}* and *Pou1f1-Cre;Rosa26^{flSTOPNICD2/+}* 6 week-old pituitaries in males and females (*Figure 3C,D*). This GH deficiency remains to a similar extent in 14-week-old *Pou1f1-Cre;Rosa26^{flSTOPNICD1/+}* male mice, whilst becoming more severe in adult females. In contrast, there was no reduction in pituitary GH in 14 week-old *Pou1f1-Cre;Rosa26^{flSTOPNICD2/+}* animals (*Figure 3C*).

We then examined the fate of recombined cells by immunofluorescence in 6 week-old animals, given that GH deficiency is present in mutants. NICD1iresnucGFP-positive cells show expression of differentiation markers such as GH, PRL and TSH suggesting that cell fate acquisition is not prevented by NOTCH ectopic activation (*Figure 3D*).

These results are in sharp contrast with a previous study where a *Pou1f1-NICD1* transgene was shown to result in severe dwarfism and blockade of cell fate acquisition in the POU1F1 lineage (*Zhu et al., 2006*). We thus decided to examine in more detail the effects of ectopic activation in the embryo. When we compared NICD1iresnucGFP expression with that of eYFP in control embryos at 18.5dpc, we observed a similar pattern of recombination with recombined cells expressing POU1F1, GH and TSH (*Figure 4A*, *Figure 4—figure supplement 1*). We did not examine lactotrophs as these mostly emerge post-natally. We then examined NOTCH pathway activation by RT-qPCR at 18.5dpc and observed a non-significant increase in *Hes5* expression (*Figure 4B*), in contrast with the robust up-regulation observed in adults (*Figure 3B*). *Heyl* and *Hey1* expression levels are significantly up-



**Figure 3.** Activation of NOTCH pathway in POU1F1 lineage alters somatotrophs post-natally. (**A**) Growth curves of *Pou1f1-Cre; Rosa26*<sup>flSTOPNICD1/+</sup> and *Pou1f1-Cre; Rosa26*<sup>flSTOPNICD2/+</sup> animals. Male (M) and female (F) mutant (T) and wild-type (NT) littermates were monitored from weaning at 3 weeks of age until 14 weeks of age. There is a transient significant weight reduction around puberty in *Pou1f1-1Cre;R26*<sup>flSTOPNICD2/+</sup> animals. A similar tendency is observed in *Pou1f1-1Cre;R26*<sup>flSTOPNICD1/+</sup> animals but this does not reach significance. (n = 3 to 14 mice/sex and genotype, mixed-effects model of

*Figure 3 continued on next page*

**Figure 3 continued**

weight versus age was used, using genotype as fixed factors and subject (mice) as random factors, with analysis of variance (ANOVA) to test the overall effect of genotype on growth, followed by Tukey post-hoc tests). (B) RT-qPCR analysis of NOTCH target genes in *Pou1f1-1Cre;R26*$^{flSTOPNICD1\ and\ 2/+}$ adult pituitaries. There is a significant increase in expression levels of three NOTCH targets, *Hes1*, *Hes5* and *Heyl* in *Pou1f1-1Cre;R26*$^{flSTOPNICD1/+}$ animals and *Hes1*, *Hes5* and *Hey1* in *Pou1f1-1Cre;R26*$^{flSTOPNICD2/+}$ animals. This demonstrates that the pathway is overactivated (n = 3 to 6 pituitaries/ genotype in *Pou1f1-Cre; Rosa26*$^{flSTOPNICD1/+}$ panel –NT- are *Rosa26*$^{flSTOPNICD1/+}$ and –T- *Pou1f1-Cre; Rosa26*$^{flSTOPNICD1/+}$ unpaired t test performed). (C) Growth hormone (GH) pituitary contents were assayed by sandwich ELISA. GH contents of *Pou1f1-1Cre;R26*$^{flSTOPNICD1/+}$ male (p=0.0003 at 6 weeks, p=0.0032 at 14 weeks) and female (p=0.0048 at 6 weeks and p=0.0013 at 14 weeks) mice are significantly reduced at both 6 and 14 weeks of age compared to wild-type littermates. In *Pou1f1-1Cre;R26*$^{flSTOPNICD2/+}$ there is only a transient deficit at 6week-old (p<0.0001 for males and p=0.0219 for females). This shows that somatotrophs function is affected by NICD expression (n = 3 to 6 mice/sex and genotype, unpaired t test performed). (D) Immunofluorescence of 6 week-old *Pou1f1-1Cre;R26*$^{flSTOPNICD1/+}$ pituitaries. NICD1iresGFP positive cells show expression of GH, PRL and TSH (with high magnification inset), suggesting that somatotroph, lactotroph and thyrotroph cell fate acquisition has not been impaired by NOTCH activation. Scale bar represent 50 µm.

DOI: https://doi.org/10.7554/eLife.33318.009

The following source data is available for figure 3:

**Source data 1.** Weights for Pou1f1Cre;R26 flSTOPNICD1/+ growth curves.
DOI: https://doi.org/10.7554/eLife.33318.010
**Source data 2.** Weights for Pou1f1Cre;R26 flSTOPNICD2/+ growth curves.
DOI: https://doi.org/10.7554/eLife.33318.011
**Source data 3.** GH contents for Pou1f1Cre;R26 flSTOPNICD1/+.
DOI: https://doi.org/10.7554/eLife.33318.012
**Source data 4.** GH contents for Pou1f1Cre;R26 flSTOPNICD2/+.
DOI: https://doi.org/10.7554/eLife.33318.013

regulated but they are still well below activation levels observed post-natally for *Hes5*. In addition, *Pou1f1* expression levels are not affected in mutants.

In summary, ectopic expression of NICD in POU1F1-positive cells does not initially induce an efficient activation of the NOTCH pathway. This suggests that in the embryo POU1F1-positive cells are resistant to NOTCH activation. In consequence, cell fate acquisition is not affected and cells apparently differentiate normally. In adult animals, however, the pathway ultimately becomes activated and this correlates with an impairment of somatotroph function.

## Loss of the NOTCH signalling mediator RBPj in the POU1F1 lineage does not significantly affect endocrine function

We then investigated the consequences of inhibiting signalling transduced through all NOTCH receptors, by conditional deletion of the NOTCH signalling mediator *Rbpj*. *Pou1f1-Cre;Rbpj*$^{fl/fl}$ mice were born and they did not present any obvious phenotype. Growth curves did not reveal any significant weight differences from control littermates (*Figure 5A*). Furthermore, total pituitary GH contents of male and female *Pou1f1-Cre;Rbpj*$^{fl/fl}$ mice at 6 and 14 weeks of age do not show any significant changes (*Figure 5B,C*). Altogether these results suggest that physiologically, the classical NOTCH signalling pathway does not play a significant role for somatotroph emergence and function post-natally.

## Activation of NOTCH pathway in the POMC lineage does not prevent emergence of endocrine cells, but results in a progressive downregulation of differentiation markers

Because POU1F1-positive cells appeared resistant to ectopic NOTCH activation, we wondered whether this was true for other endocrine lineages. We therefore decided to investigate the effect of NOTCH ectopic activation in the POMC lineage. It has previously been shown that ectopic activation in this lineage using the same *Rosa26*$^{flSTOPNICD1}$ allele results in loss of corticotrophs and melanotrophs (*Goldberg et al., 2011*). Here, we used a different transgenic POMC-Cre exclusively active in the pituitary, in essentially all melanotrophs, and half of corticotrophs (*Langlais et al., 2013*). We have previously characterised this POMC-Cre and observed that the pattern of recombination closely matches expression of the endogenous gene (*Goldsmith et al., 2016*).



**Figure 4.** Expression of NICD1 in POU1F1 lineage does not alter cell fate acquisition in the developing pituitary and this correlates with inefficient activation of NOTCH pathway. (**A**) Immunofluorescence of 18.5dpc *Pou1f1-Cre;Rosa26*<sup>ReYFP</sup> and *Pou1f1-Cre;Rosa26*<sup>flSTOPNICD1/+</sup>pituitaries. In both control and mutant embryos, the pattern of recombination appears similar. EYFP, in controls, and NICD1iresGFP positive cells in mutants co-express PIT1, GH or TSH, suggesting that cell fate acquisition is not altered. Scale bars represent 50 µm (upper panels, low magnification) and 10 µm (lower

*Figure 4 continued on next page*

Figure 4 continued
panels, higher magnification). (B) RT-qPCR analysis of NOTCH target genes in *Pou1f1-1Cre;R26^flSTOPNICD1/+* pituitaries at 18.5dpc. We only observe a significant, but mild upregulation of *Hey1* and *Heyl* in *Pou1f1-1Cre;R26^flSTOPNICD1/+* pituitaries compared to both *Pou1f1-1Cre* and *R26^flSTOPNICD1/+* controls (n = 5 to 10 pituitaries/genotype, unpaired t test performed). Consequently, *Pou1f1* expression levels are not affected in mutants. This suggests that the pathway cannot be efficiently activated in the POU1F1 lineage.
DOI: https://doi.org/10.7554/eLife.33318.014
The following figure supplement is available for figure 4:

**Figure supplement 1.** Characterization of POU1F1-Cre by lineage tracing analysis at 18.5dpc.
DOI: https://doi.org/10.7554/eLife.33318.015

As previously described, we compared patterns of eYFP and NICDiresnucGFP expression in respectively *Pomc-Cre;Rosa26^ReYFP/+* and *Pomc-Cre;Rosa26^flSTOPNICD1/+* at 18.5dpc (*Figure 6A*). Patterns of recombination appear similar. Moreover, we observe in both genotypes co-localisation of the fluorescent protein with POMC, both in AL corticotrophs and IL melanotrophs. Co-localisation with PAX7 in IL demonstrates that melanotroph emergence occurs normally (*Figure 6A*).

In contrast with what was observed in *Pou1f1-Cre;Rosa26^flSTOPNICD1/+* embryos (*Figure 4B*), quantification revealed a much more significant up-regulation of several NOTCH pathway target genes in 16.5 and 18.5dpc mutant pituitaries (*Figure 6B*), comparable with what we measured in *Nkx3.1^Cre/+*; *Rosa26^floxSTOPNICD1 and 2/+* embryos (*Figure 1A*). We examined *Hes5* expression levels in more detail, because it is the most robustly induced target, and observed a two-fold increase between 16.5dpc and 18.5dpc (*Figure 6C*). This may reflect induction in melanotrophs, because these only start to express POMC at 16.5dpc; however there may also be an increase in activation in corticotrophs. We then examined expression levels of factors required for POMC lineage emergence and differentiation (*Figure 6D,E*). Despite observing an apparent equivalent level of POMC staining by immunofluorescence (*Figure 6A*), we detected a significant down-regulation of the gene expression at 18.5dpc (*Figure 6C*). Expression of the transcription factor *Tbx19*, encoding a direct activator of *Pomc* expression (*Lamolet et al., 2001*), and *Ascl1*, encoding for a basic helix loop helix (bHLH) transcription factor expressed in the POMC lineage (*Liu et al., 2001*) were both significantly reduced. In contrast, expression of *Neurod1*, another bHLH factor transiently required for corticotroph differentiation (*Lamolet et al., 2004*), was not affected. In melanotrophs, expression of *Pax7* was not significantly affected. In contrast, expression of the SC marker *Sox9* was significantly upregulated. Immunofluorescence for SOX9 at 18.5dpc shows a robust ectopic expression of the protein in melanotrophs at 18.5dpc (*Figure 6F*). SOX2 is normally expressed in melanotrophs, but at lower levels than in SC (*Goldsmith et al., 2016*). Immunofluorescence for SOX2 in *Pomc-Cre;Rosa26^flSTOPNICD1/+* melanotrophs suggested that levels were elevated and comparable to what we observe in stem cells (*Figure 6—figure supplement 1*), but gene expression was not significantly altered in RT-qPCR assays (data not shown). In AL, SOX2 and SOX9 are not up-regulated in NICD1iresGFP corticotrophs.

Because POMC staining appeared normal (*Figure 6A*), while expression of the gene was significantly downregulated at 18.5dpc (*Figure 6D*), this suggested that NOTCH activation was progressively affecting the cells, and *Pomc* expression was gradually down-regulated. We therefore examined expression of *Pomc*, *Tbx19* and *Neurod1* at 16.5dpc (*Figure 6E*). We observe a significant down-regulation of *Pomc* and *Tbx19* (respectively 50% and 65% decrease compared to Cre positive samples), but this was not as dramatic as that observed at 18.5dpc (respectively 78% and 81% decrease compared to Cre positive samples), strongly suggesting a progressive impairment of differentiation as NOTCH activation becomes stronger (*Figure 6D*).

In summary, ectopic NOTCH activation does not appear to prevent emergence of corticotrophs and melanotrophs, because POMC is clearly present in both cell types at 18.5dpc (*Figure 6A*). However, expression of both the hormone precursor, and its direct activator, *Tbx19*, are dramatically downregulated during late gestation. We propose that sufficient levels of TBX19 during early development, and normal levels of NEUROD1 may allow emergence of POMC positive cells (*Figure 6D, E*). In parallel, we observe ectopic expression of SOX9 and slightly increased levels of SOX2, two pituitary SC markers, in melanotrophs.

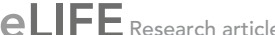

**Figure 5.** Deletion of NOTCH pathway transcriptional mediator *Rbpj* in POU1F1 lineage has no effect on GH levels. (**A**) Growth curves of *Pou1f1-Cre;Rbpj*<sup>fl/fl</sup> animals. Male (M) and female (F) mutant (T) and wild-type (NT) littermates were monitored from weaning at 3 weeks of age until 14 weeks of age. There is no significant difference between wild-type littermate and mutants (n = 3 to 9 mice/sex and genotype, mixed-effects model of weight versus age was used, using genotype as fixed factors and subject (mice) as random factors, with analysis of variance (ANOVA) to test the overall effect of genotype on growth, followed by Tukey post-hoc tests). (**B, C**) GH
*Figure 5 continued on next page*

*Figure 5 continued*

pituitary contents were assayed by sandwitch ELISA in 6 week-old (B, n = 3 to 6 pituitaries/sex and genotype) and 14 week-old (C, n = 3 to 7 pituitaries/sex and genotype) animals. There is no significant difference between wild-type and mutant animals (unpaired t test performed).

DOI: https://doi.org/10.7554/eLife.33318.016

The following source data is available for figure 5:

**Source data 1.** Weights for Pou1f1Cre;RBPJfl/fl growth curves.

DOI: https://doi.org/10.7554/eLife.33318.017

**Source data 2.** GH contents for Pou1f1Cre;RBPJfl/fl 42 days-old.

DOI: https://doi.org/10.7554/eLife.33318.018

**Source data 3.** GH contents for Pou1f1Cre;RBPJfl/fl 100 days-old.

DOI: https://doi.org/10.7554/eLife.33318.019

## Activation of NOTCH pathway in the POMC lineage induces post-natal regression of melanotrophs and corticotrophs to a progenitor-like state

*Pomc-Cre;Rosa26$^{flSTOPNICD1/+}$* animals are born and viable. In contrast with the POMC staining observed at 18.5dpc in mutants (*Figure 6A*), but in agreement with a dramatic down-regulation of the gene expression (*Figure 6D*), we observe early post-natally a spectacular reduction of POMC staining in both IL and AL NICD1iresGFP-positive cells (*Figure 7A*). In 4 month-old animals (*Figure 7B–E*), SOX9 is still ectopically and robustly expressed in the POMC negative, PAX7;SOX2 positive IL cells (*Figure 7B,C*). In contrast, in AL mutant corticotrophs, SOX2, but not SOX9, is ectopically up-regulated (*Figure 7D,E*). We examined the expression of P57, a marker of post-mitotic undifferentiated progenitors, and P27 whose expression replaces that of P57 in differentiated cells (*Bilodeau et al., 2009*). While expression of P27 remains similar to controls in mutant IL, a few cells re-activated expression of P57, in agreement with regression toward a progenitor state (*Figure 7— figure supplement 1*).

Mutant ILs clearly appear thinner, however it is also noticeable that cell density is higher compared to controls. To understand the origin of this difference, we initially examined cell apoptosis at P2 but did not observe any TUNEL-positive staining in NICD1iresGFP-positive cells (*Figure 7—figure supplement 2*). We then examined cell proliferation at the same stage by quantifying EdU/PAX7 double positive cells and observed a significant increase in EdU incorporation in *Pomc-Cre;Rosa26$^{fl-STOPNICD1/+}$* animals (*Figure 7F*). This is in agreement with induction of a progenitor-like fate, characterised at these early post-natal stages by a much higher proliferative capacity compared to differentiated cells (*Gremeaux et al., 2012*). We then quantified cell density in 5 month-old ILs and observe a significant increase in mutants (*Figure 7G*), strongly suggesting that the reduction in IL size is not due to reduced cell numbers, but to reduced cell volume, in agreement with a progenitor-like identity of mutant cells.

Induction of *Sox9* in response to NOTCH activation has previously been reported, and shown to be of functional significance (*Martini et al., 2013*). We therefore investigated whether deletion of *Sox9* in *Pomc-Cre;Rosa26$^{flSTOPNICD1/+}$* could rescue the IL regression phenotype. We did not observe an increase in POMC staining or IL thickness in 3 week-old *Pomc-Cre;Rosa26$^{flSTOPNICD1/+}$;Sox9$^{fl/fl}$* samples (*Figure 7H*), demonstrating that SOX9 is not a mediator of NICD1 activation, at least for these effects.

In summary, these data suggest that NOTCH activation results in a regression of POMC lineage cells toward a stable progenitor-like state, but with different outcomes in IL (SOX2; SOX9 double positive) and AL (SOX2 positive) cells.

## Discussion

To better characterize the role of the NOTCH pathway during pituitary development, we have manipulated its activity in a comparable manner in different cellular contexts. Our experiments reveal an unexpected differential sensitivity to NICD expression between lineages and, in consequence, provide a better understanding of the physiological role of the pathway during pituitary development. Progenitors, normally exposed and responsive to NOTCH signalling, are sensitive to



**Figure 6.** Activation of NOTCH pathway in the POMC lineage results in a gradual loss of differentiation markers in the embryo. (**A**) Immunofluorescence of 18.5dpc *Pomc-Cre;Rosa26*$^{ReYFP}$ and *Pomc-Cre;Rosa26*$^{flSTOPNICD1/+}$ pituitaries. In both control *Pomc-Cre;Rosa26*$^{ReYFP}$ and mutant *Pomc-Cre;Rosa26*$^{flSTOPNICD1/+}$ embryos, the pattern of recombination appears similar. EYFP, in controls, and NICD1iresGFP positive cells in mutants co-express PAX7 and POMC (with high magnification inset), suggesting that cell fate acquisition is not altered. Fluorescence detection thresholds are

*Figure 6 continued on next page*

*Figure 6 continued*

identical between control and mutant. (B) RT-qPCR analysis of NOTCH pathway target genes in 18.5dpc pituitaries. There is a significant induction of *Hes5*, *Heyl*, and *Hey1* expression following NICD expression (n = 3 to 4 pituitaries/genotype unpaired t test performed). (C) RT-qPCR analysis of *Hes5* in 16.5 and 18.5dpc *Pomc-Cre;Rosa26*<sup>flSTOPNICD1/+</sup> pituitaries. Levels of *Hes5* are doubling between 16.5 and 18.5dpc (n = 4 to 7 pituitaries/genotype, unpaired t test performed). (D) RT-qPCR analysis of POMC lineage cell-type markers in 18.5dpc *Pomc-Cre;Rosa26*<sup>flSTOPNICD1/+</sup> pituitaries. There is a significant reduction of *POMC*, *Tbx19* and *Ascl1* expression while *Sox9* is up-regulated. In contrast expression of *Pax7* and *Neurod1* is not significantly affected (n = 4 to 6 pituitaries/genotype, unpaired t test performed). (E) RT-qPCR analysis of POMC lineage cell-type markers in 16.5dpc *Pomc-Cre; Rosa26*<sup>flSTOPNICD1/+</sup> pituitaries. The same tendency is observed, but downregulation is milder, suggesting that gene expression is further altered as *Hes5* induction becomes more robust (n = 5 to 7 pituitaries/genotype, unpaired t test performed). (F) Immunofluorescence of 18.5dpc *Rosa26*<sup>flSTOPNICD1/+</sup> and *Pomc-Cre;Rosa26*<sup>flSTOPNICD1/+</sup> pituitaries. In the control *Rosa26*<sup>flSTOPNICD1/+</sup> section, PAX7 is present in melanotrophs, while SOX9 is restricted to stem cells lining the cleft in IL and AL, and glial cells in PL. PAX7 and SOX9 are hardly expressed by the same cells. In contrast, in *Pomc-Cre;Rosa26*<sup>flSTOPNICD1/+</sup> there is a clear co-localisation of SOX9 and PAX7 in IL. Scale bar represents 50 μm. IL is underlined.
DOI: https://doi.org/10.7554/eLife.33318.020

The following figure supplement is available for figure 6:

**Figure supplement 1.** In *Pomc-Cre;Rosa26*<sup>flSTOPNICD1/+</sup> SOX2 expression levels appear similar in progenitors and melanotrophs.
DOI: https://doi.org/10.7554/eLife.33318.021

its activation and this results in a blockade of cell fate acquisition. However, we show here that timing and/or duration of activation in progenitors differentially affects cell fate acquisition and maintenance in the prospective IL. Strikingly, in differentiating endocrine cells, we then uncover that NOTCH ectopic activation induces a blunted response in the POU1F1 lineage, while POMC-positive cells remain sensitive to its effect. Consequently, cell fate acquisition and maintenance of differentiation in POU1F1-positive cells is apparently not affected, suggesting that the NOTCH pathway does not play a role for this process in this lineage. In agreement, deletion of the NOTCH effector RBPJ has no apparent phenotypic consequence in this lineage. In contrast, in the POMC lineage, NOTCH activation results in a gradual dedifferentiation of both corticotrophs and melanotrophs toward a progenitor-like state. Taken together these results demonstrate first that the NOTCH pathway maintains progenitors in an undifferentiated state, however, response to activation is modulated in time in dorsal progenitors. Second, sensitivity to NOTCH signalling in the developing pituitary varies in the different endocrine lineages. We suggest that the maintenance of high responsiveness to NOTCH activation in the POMC lineage, but not in the POU1F1 cells, points to a physiological requirement for the NOTCH pathway to specifically prevent corticotroph and melanotroph cell differentiation.

## Role of NOTCH pathway in progenitors.

When NOTCH signalling is prevented in pituitary progenitors, either by deleting *RBPJ* (**Zhu et al., 2015**; **Zhu et al., 2006**), the target genes *Hes1* and *Hes5* (**Kita et al., 2007**; **Raetzman et al., 2007**), or the *Notch2* receptor (**Nantie et al., 2014**), reduction of cell proliferation and precocious differentiation are observed. Conversely, ectopic expression of *Hes1* in both gonadotrophs and thyrotrophs blocks their differentiation (**Raetzman et al., 2007**), and ectopic expression of *Notch2* in the same cells delays differentiation and maturation (**Raetzman et al., 2006**). This suggests that NOTCH signalling maintains progenitors in an undifferentiated state and that downregulation of its activity is required for cells to differentiate, as observed in other tissues (for review see (**Bray, 2016**)). In agreement with these studies (**Kita et al., 2007**; **Raetzman et al., 2007**; **Zhu et al., 2015**; **Zhu et al., 2006**);(**Nantie et al., 2014**), we show here that maintained activation of the NOTCH pathway in pituitary progenitors efficiently blocks cell fate acquisition. However, progression of progenitors, exemplified in the pituitary by the transition from an early SOX2-positive;SOX9-negative highly proliferative cell to a SOX2;SOX9 double-positive more quiescent progenitor/SC at late gestation (**Rizzoti et al., 2016**), does not appear to be prevented by NOTCH activity. This is because we still observe up-regulation of SOX9 expression in *Sox2*<sup>CreERT2</sup>;*Rosa26*<sup>floxSTOP-NICD1</sup> progenitors. Therefore, NOTCH specifically affects cell fate acquisition, but apparently not other aspects of progenitor identity and fate.

Blockade of cell fate acquisition appears more efficient in *Sox2*<sup>CreERT2</sup>;*Rosa26*<sup>floxSTOP-NICD1</sup> progenitors induced at 9.5dpc than in *Nkx3.1*<sup>Cre</sup>;*Rosa26*<sup>floxSTOP-NICD1</sup> ones, because the proportion of recombined cells that retain SOX2 expression is significantly higher in the former samples. In RP,



**Figure 7.** Activation of NOTCH pathway in the POMC lineage results in regression of melanotrophs and corticotrophs toward a progenitor-like fate post-natally. (**A**) Immunofluorescence of P7 wild-type and *Pomc-Cre;Rosa26^{flSTOPNICD1/+}* pituitaries. Expression of POMC is dramatically down-regulated in IL and NICD1iresGFP positive corticotrophs in AL. SOX2 staining appears of similar intensity in both IL and SCs in mutants, while in control its expression is clearly lower in IL melanotrophs compared to SCs. (**B, C**) Immunofluorescence of 4 month-old wild-type and *Pomc-Cre;Rosa26^{flSTOPNICD1/+}* IL. There is a clear co-localisation of PAX7 and SOX9 in mutant IL that is never observed in wild-type samples (**B**). POMC expression is almost absent in the SOX2 positive IL cells in mutants (**C**). In addition, IL is thinner in mutants, while cell density appears increased. (**D, E**) Immunofluorescence of 4 month-old wild-type and *Pomc-Cre;Rosa26^{flSTOPNICD1/+}* AL. In contrast with what is observed in IL, SOX9 is not expressed in AL NICD1iresGFP positive cells (**D**), but SOX2 is ectopically induced in the mutant cells (arrow, **E**). (**F**) Analysis of cell proliferation in PAX7 positive cells at P2. Pups were injected with EdU and pituitaries harvested one hour later. There is a significant increase in cell proliferation in *Pomc-Cre;Rosa26^{flSTOPNICD1/+}* PAX7 positive cells

*Figure 7 continued on next page*

*Figure 7 continued*

(n = 3 to 4 pups/genotype, unpaired t test performed, p=0.0013 is calculated after angular transformation of percentages). (**G**) Analysis of cell density in 5 month-old IL (n = 3 to 4 pituitaries/genotype, unpaired t test performed, p=0.0011). (**H**) Immunofluorescence for SOX9 and POMC in 3 week-old pituitaries. Despite loss of ectopic SOX9 expression in *Pomc-Cre;Rosa26*$^{flSTOPNICD1/+}$;*Sox9* $^{fl/fl}$ POMC expression is still down-regulated in the hypoplastic IL. Scale bar represent 50 μm in A, H and in B for C-E. IL is underlined.
DOI: https://doi.org/10.7554/eLife.33318.022

The following source data and figure supplements are available for figure 7:

**Source data 1.** Countings for graph F.
DOI: https://doi.org/10.7554/eLife.33318.025
**Source data 2.** Countings for graph G.
DOI: https://doi.org/10.7554/eLife.33318.026
**Figure supplement 1.** P57 is up-regulated in some cells in *Pomc-Cre;Rosa26*$^{flSTOPNICD1/+}$ IL.
DOI: https://doi.org/10.7554/eLife.33318.023
**Figure supplement 2.** Reduction of intermediate lobe size in *Pomc-Cre;Rosa26*$^{flSTOPNICD1/+}$ pituitaries is not due to cell apoptosis.
DOI: https://doi.org/10.7554/eLife.33318.024

*Nkx3.1* is initially expressed at 10.5dpc, in a mosaic pattern, then up-regulated in the dorsal region of RP at 12.5 until at least 14.5dpc (*Treier et al., 2001*). This domain encompasses that of SOX2 expression, and 70% of SOX2-positive cells are also EYFP-positive in 12.5dpc *Nkx3.1*$^{Cre}$;*Rosa26*$^{ReYFP}$ RPs (S. Goldsmith, PhD thesis). It is possible that some cells where *Nkx3.1*$^{Cre}$ is active are not progenitors since we have not established co-localisation between NKX3.1 and SOX2 at all stages; this would explain why fewer cells express SOX2 in *Nkx3.1*$^{Cre}$;*Rosa26*$^{floxSTOP-NICD1}$ compared to *Sox2-*$^{CreERT2}$;*Rosa26*$^{floxSTOP-NICD1}$ pituitaries. However, it is likely that the earlier NOTCH activation is induced, the more efficiently cell fate acquisition is prevented. This implies that, as progenitors progress toward commitment, some become less sensitive to its effects, perhaps those up-regulating POU1F1 expression (see below), because we do observe some rare cells where SOX2 and POU1F1 co-localise (*Fauquier et al., 2008*).

The differential sensitivity of progenitors is also well illustrated by the contrasting effects of NOTCH activation in IL progenitors. Indeed, in *Sox2*$^{CreERT2}$;*Rosa26*$^{floxSTOP-NICD1}$ embryos we observe an exclusion of recombined cells from the future IL. This exclusion appears to be an active process because recombined cells lose some epithelial characteristics and appear to integrate into the AL. This process is reminiscent of a phenomenon recently described in *Drosophila* where juxtaposition, within an epithelium, of cells with different fates results in extrusion of single cells (*Bielmeier et al., 2016*). A similar mechanism could operate here, where recombined cells with higher levels of NOTCH activity are singled-out because they are unable to commit to a dorsal/IL fate. Alternatively, or in addition, NOTCH signalling activity is polarized in epithelial cells (*Perez-Mockus and Schweisguth, 2017*). It is therefore possible that our NICD expression approach directly affects epithelial properties. Surprisingly, however, this effect is transient, because activation of the NOTCH pathway 72 hr later, using either *Nkx3.1*$^{Cre}$ or inducing *Sox2*$^{CreERT2}$ at 12.5dpc, does not result in cell extrusion. Instead, as we observe in AL, cell fate acquisition is mostly prevented, with some rare cells committing to a PAX7 positive melanotroph fate, but unable to progress to differentiation and to up-regulate POMC expression. These data suggest that between 9.5 and 12.5 dpc a dorsal fate may be imparted to cells in the prospective IL, and that this regionalisation information is sensitive to NOTCH signalling.

## NOTCH signalling in POU1F1 and POMC lineages.

The effects of NICD ectopic expression in the POU1F1 (*Zhu et al., 2006*) and POMC lineages (*Goldberg et al., 2011*) have previously been reported. However, both the approaches used and the outcomes are different from ours. Zhu et al engineered a POU1F1-NICD transgene and observe a dramatic blockade of cell differentiation (*Zhu et al., 2006*), which is in sharp contrast with our observations. This discrepancy may be explained by an earlier and maybe higher expression of the POU1F1-NICD transgene compared to NICD in our *Pou1f1-Cre;Rosa26*$^{flSTOPNICD1/+}$ samples. If the transgene was expressed in cells that are not yet committed, then blockade of cell differentiation would be expected. Similarly, in the *Pomc-Cre;Rosa26*$^{flSTOPNICD1/+}$ embryos described in Goldberg et al, POMC expression is never detected, suggesting that corticotrophs and melanotrophs do not

emerge, again in contrast with our results. This discrepancy could be explained by an earlier expression of the *Pomc-cre* transgene (*Balthasar et al., 2004*) used in Goldberg et al, which is different from the one used here (*Langlais et al., 2013*). We have previously characterised this Cre transgene (*Langlais et al., 2013*) in the embryonic pituitary, and showed that it matches endogenous POMC expression (*Goldsmith et al., 2016*).

Consistently, our results show that neither ectopic expression of NICD, nor deletion of RBPJ in the POU1F1 lineage affect cell fate acquisition. In agreement with an apparent absence of a phenotypic effect, we show that ectopic expression of NICD hardly affects expression of NOTCH target genes in the embryonic pituitary, demonstrating that the cells are relatively insensitive to its effect. This lack of responsiveness may be explained by sequestration of NOTCH responsive enhancers by other factors, as observed in different contexts (for review see (*Bray, 2016*)). Therefore, the classical NOTCH pathway is unlikely to be directly involved for POU1F1 lineage commitment and differentiation. However, there is a significant activation of the pathway by NICD post-natally and a reduction in GH production and secretion, therefore, the NOTCH pathway activation compromises some aspects of somatotroph function.

In contrast, in the POMC lineage, responsiveness to NICD expression is comparable to what is observed in progenitors. Following activation, expression of *Tbx19* (*Lamolet et al., 2001*), the key determinant of melanotroph and corticotroph identity, is gradually downregulated in the embryo. However, expression of *Neurod1* (*Lamolet et al., 2004*), transiently required for early corticotroph differentiation, is not significantly affected. Both TBX19 and NEUROD1 can activate the expression of POMC (*Lamolet et al., 2001*) (*Poulin et al., 1997*). We therefore propose that unaffected levels of NEUROD1 and sufficient levels of TBX19 in early stages of terminal differentiation allow up-regulation of POMC expression in *Pomc-Cre;Rosa26^{flSTOPNICD1/+}* corticotrophs. Corticotrophs are the first pituitary endocrine cells to up-regulate expression of the hormone they secrete. Our results point to a direct and specific effect of NOTCH signalling in preventing their differentiation, presumably through down-regulation of *Tbx19*. This is in agreement with results obtained in pituitary organoids (*Suga et al., 2011*) where efficient and specific generation of corticotrophs is observed after inhibition of NOTCH pathway. In IL, expression of the melanotroph pioneer factor *Pax7* is not affected, therefore cell fate acquisition is not impaired by NOTCH activation. However, as observed in corticotrophs, gradual downregulation of *Tbx19* results in loss of POMC expression post-natally.

Regression of both IL and AL in cells with *Pomc-Cre;Rosa26^{flSTOPNICD1/+}* is associated with up-regulation of the pituitary SC markers SOX2 and SOX9 (*Rizzoti et al., 2013*). This suggests that the cells partially dedifferentiate. In agreement with this hypothesis, IL mutant cells proliferate more, at least early post-natally. We observe a differential expression of the two SOX transcription factors post-natally: SOX9 is strongly up-regulated and maintained in *Pomc-Cre;Rosa26^{flSTOPNICD1/+}* IL but not in AL. In contrast SOX2, normally present in melanotrophs (*Goldsmith et al., 2016*), is up-regulated in *Pomc-Cre;Rosa26^{flSTOPNICD1/+}* corticotrophs. Both factors belong to different SOX families therefore, they do not perform redundant functions. In addition, in different cellular contexts, expression of SOX2 (*Pan et al., 2013*) or SOX9 (*Martini et al., 2013*) is induced in response to NOTCH activation, and this is required for NOTCH-dependant progenitor maintenance. The significance of this differential expression is unclear. However we show here that deletion of *Sox9* in *Pomc-Cre;Rosa26^{flSTOPNICD1/+}* pituitaries does not prevent IL regression, suggesting that SOX9 does not play a significant role in the phenotype. Further analysis is necessary to better characterise the role of these factors following NOTCH activation in the POMC lineage.

In conclusion, the regression of both melanotrophs and corticotrophs observed in response to NICD expression shows that the NOTCH pathway most likely has a direct role in preventing POMC lineage differentiation. In addition, the lasting sensitivity of this lineage to NOTCH activation and its dedifferentiating effect may be of relevance in pathological situations, such as tumorigenesis. This also raises the possibility that this lineage may be more easily reprogrammed, potentially to promote regeneration. It is tempting to speculate that this specificity reflects the absolute requirement for a functioning hypothalamo-pituitary-adrenal axis for survival. Further characterization of NICD action in this lineage is now essential.

# Materials and methods

## Key resources table

| Reagent type (species) or resource | Designation | Source or reference | Identifiers | Additional information |
|---|---|---|---|---|
| strain, strain background (*mus musculus*) | *Nkx3.1$^{Cre}$* | Lin, Y., Liu, G., Zhang, Y., Hu, Y.P., Yu, K., Lin, C., McKeehan, K., Xuan, J.W., Ornitz, D.M., Shen, M.M., *et al.* (2007). Fibroblast growth factor receptor 2 tyrosine kinase is required for prostatic morphogenesis and the acquisition of strict androgen dependency for adult tissue homeostasis. Development *134*, 723–734. | Nkx3-1$^{tm3(cre)Mms}$ | |
| strain, strain background (mus musculus) | *Sox2$^{CreERT2}$* | Arnold, K., Sarkar, A., Yram, M.A., Polo, J.M., Bronson, R., Sengupta, S., Seandel, M., Geijsen, N., and Hochedlinger, K. (2011). Sox2(+) adult stem and progenitor cells are important for tissue regeneration and survival of mice. Cell Stem Cell *9*, 317–329. | *Sox2$^{tm1(cre/ERT2)Hoch}$* | |
| strain, strain background (mus musculus) | *Pou1f1-Cre* | this paper | *(nogene)$^{Tg(pou1f1-cre)1Rsd}$* | |
| strain, strain background (mus musculus) | *Rbpj$^{fl}$* | Han, H., Tanigaki, K., Yamamoto, N., Kuroda, K., Yoshimoto, M., Nakahata, T., Ikuta, K., and Honjo, T. (2002). Inducible gene knockout of transcription factor recombination signal binding protein-J reveals its essential role in T versus B lineage decision. Int Immunol *14*, 637–645. | RBPJ$^{tm1Hon}$ | |
| strain, strain background (mus musculus) | *Rosa26$^{ReYFP}$* | Srinivas, S., Watanabe, T., Lin, C.S., William, C.M., Tanabe, Y., Jessell, T.M., and Costantini, F. (2001). Cre reporter strains produced by targeted insertion of EYFP and ECFP into the ROSA26 locus. BMC Dev Biol *1*, 4. | *Gt(Rosa26)Sortm1(EYFP)Cos* | |
| strain, strain background (mus musculus) | *Rosa26$^{floxSTOP-Nicd1}$* | Murtaugh, L.C., Stanger, B.Z., Kwan, K.M., and Melton, D.A. (2003). Notch signaling controls multiple steps of pancreatic differentiation. Proc Natl Acad Sci U S A *100*, 14920–14925. | *Gt(Rosa26)Sortm1(Notch1)Dam* | |
| strain, strain background (mus musculus) | *Rosa26$^{floxSTOP-Nicd2}$* | Fujimura, S., Jiang, Q., Kobayashi, C., and Nishinakamura, R. (2010). Notch2 activation in the embryonic kidney depletes nephron progenitors. J Am Soc Nephrol *21*, 803–810. | *Gt(Rosa)26Sortm1(Notch2)Nis* | |
| strain, strain background (mus musculus) | *Pomc-Cre* | Langlais, D., Couture, C., Kmita, M., and Drouin, J. (2013). Adult pituitary cell maintenance: lineage-specific contribution of self-duplication. Mol Endocrinol *27*, 1103–1112. | | |
| Antibody | goat anti-Sox2 | ISS | GT15098 | 1/300 |
| Antibody | rat anti-GFP | Fine Chemical products | 04404–84 | 1/1000 |
| Antibodies | anti pituitary hormones | NHPP | | |
| Antibody | mouse anti-Pax7 | DSHB | P3U1 | 1/100 |

## Mice

*Nkx3.1$^{Cre}$* (Nkx3-1$^{tm3(cre)Mms}$ (**Lin et al., 2007**), *Sox2$^{CreERT2}$* (*Sox2$^{tm1(cre/ERT2)Hoch}$* (**Arnold et al., 2011**), *Pou1f1-Cre* ((nogene)$^{Tg(pou1f1-cre)1Rsd}$), *Rbpj$^{fl}$* (RBPJ$^{tm1Hon}$ (**Han et al., 2002**), *Rosa26$^{ReYFP}$* (Gt(Rosa26)$^{Sortm1(EYFP)Cos}$ (**Srinivas et al., 2001**), *Rosa26$^{floxSTOP-Nicd1}$* (Gt(Rosa26)$^{Sortm1(Notch1)Dam}$ (**Murtaugh et al., 2003**), *Rosa26$^{floxSTOP-Nicd2}$*(Gt(Rosa)$^{26Sortm1(Notch2)Nis}$ (**Fujimura et al., 2010**) and *Pomc-Cre* ((nogene)$^{Tg(PomC-Cre)Dro}$ (**Langlais et al., 2013**) were maintained on mixed background.

*Pou1f1-Cre* mice were generated by first inserting the cDNA coding for NLS-CRE between rabbit beta-globin intron and beta globin polyA sequences. Tissue specific expression of CRE to the pituitary gland was obtained by inserting a 15 kb upstream promoter sequence of the *Pou1f1* gene. Finally, the transgene was cloned between insulator sequences of the beta globin gene to prevent silencing of transgene expression in vivo. Pou1f1-Cre transgenic animals were obtained by standard pronuclear injection. Tissue-specific expression of CRE to the *Pou1f1* lineage was confirmed by crossing the obtained *Pou1f1-Cre* mouse line with *Rosa26^{ReYFP}* mice. Cre activity in *Sox2^{CreERT2/+}*; *Rosa26^{floxSTOP-NICD1}* embryos was induced by a single tamoxifen treatment (0.2 mg/g/day) in pregnant females. Weights of mice were recorded weekly between age-matched littermates after weaning at 3 weeks of age.

## Immunofluorescence, EdU staining, microscopy and imaging

Immunofluorescence was performed following a previously described protocol (Cheung, 2013 #1301). Mice were perfused with 4% w/v paraformaldehyde in phosphate-buffered saline (PBS), pituitaries harvested and cryosectioned at 12 µm. Sections were blocked with blocking solution (10% v/v donkey serum in PBS/0.1% v/v Triton X-100; PBST) for 1 hr, then incubated with primary antibodies in 10% blocking solution overnight at 4°C. Primary antibodies were used at the following dilutions: rat anti-GFP (Nacalai Tesque) at 1:1000; monkey anti-rat GH (NHPP) at 1:5000; rabbit anti-POMC, GH, TSH and PRL (NHPP) at 1:500; mouse anti-POMC (Sigma) 1:1000, goat anti-POU1F1 (Santa Cruz) at 1:100, Rb anti-POU1F1 (gift from S.Rhodes, Indiana University School of Medicine) at 1: 500, Mouse anti-PAX7 (DSHB) 1:100, goat anti-SOX2 (Immune System) at 1:300, rabbit anti-SOX9 (gift from F.Poulat, Institut de Génétique Humaine, Montpellier) at 1:300. Sections were washed in PBST then incubated for 1 hr at room temperature with the corresponding anti-rat, anti-goat, anti-rabbit, or anti-human secondary antibody conjugated to Alexa-Fluor 488,–555, −594, or −647 in 10% blocking solution with 1 µM 4',6-diamino-2-phenylindole (DAPI). Cell proliferation was analysed following a one hour EdU pulse (30 µg/g body weight). Incorporation was detected using a Click-iT EdU imaging kit following the manufacturer instructions (Thermo Fisher Scientific). Sections were washed with PBST and mounted using Aqua-Poly/Mount (Polysciences, Inc., Warrington, PA, USA). Fluorescent signal was captured using a Leica SPE confocal microscope.

3D reconstruction images were generated using serial haematoxylin and eosin-stained sections from *Nkx3.1^{Cre}*; *Rosa26^{floxSTOP-NICD/+1}* and *Nkx3.1^{Cre}*; *Rosa26^{floxSTOP-NICD2/+}* embryos and controls. Sections were first imaged using an Olympus VS120 slide scanner. Images were then aligned with FIJI, and false-coloured using the ilastik 0.5 software (*Sommer et al., 2011*). Volocity software (Perkin Elmer) was used to create 3D reconstructions of these images.

## Cell countings

For EdU quantification, PAX7 positive cells and EdU positive cells were counted in three sections/embryos in three embryos/genotype. For cell density quantification, on 3 DAPI stained sections/sample (minimum three samples/genotype), nuclei were counted in a representative fixed surface area.

## mRNA extraction and reverse transcriptase-PCR (rt-qPCR)

Total mRNA was extracted from dissected embryonic and postnatal pituitary glands using the RNeasy Mini and Micro kits (Qiagen) according to the manufacturer's protocol. The extracted mRNA was transcribed into cDNA using the Quantitech Reverse Transcription kit (Qiagen, for *Figures 1* and *3*) or the Superscript VILO cDNA synthesis kit (Thermo Fisher Scientific, for *Figures 4* and *6*) according to the manufacturer's protocol.

## Quantitative real-time PCR (RT-qPCR)

Each sample was assayed in technical duplicate with each tube containing diluted template cDNA, 250 nM primers and 1xSensiMixSybr (QuantACE, *Figures 1* and *3*) or 1xAbsoluteSybrGreen ROX mix (Thermo Fisher Scientific, for *Figures 4* and *6*). Each sample was assayed for the genes of interest together with three reference housekeeping genes: *Atp5β*, *Sdha*, and *Eif1a2* (*Figures 1* and *3*) or using one reference housekeeping gene, *Gapdh* (*Figures 4* and *6*). Relative expression of the genes of interest were calculated by normalisation of the detected expression value to the geometric mean of the reference genes using the ΔΔCt method (*Bustin et al., 2009*; *Livak and Schmittgen,*

*2001*). The primers used for RT-qPCR are listed in *Supplementary file 1*. Data is shown as ±mean SEM for a minimum of 3 samples/data point.

## Sandwich enzyme-linked immunosorbent assay (ELISA)

Experimental data were collected from virgin male and female mice, unless otherwise stated. Total pituitary GH contents were assayed using a previously described method (*Steyn et al., 2011*) using mouse-specific reagents kindly provided by A.F. Parlow (National Hormone and Pituitary Program (NHPP), Torrance, CA, USA).

## Statistical analyses

Quantitative PCR, growth curves and GH assays results are presented as means ± standard error of the mean (SEM). Other quantifications are presented as means ± standard deviation (SD). When comparing two groups of values, the unpaired Student's t-test was used to produce a p-value. When repeated measurements from the same samples were taken (i.e. growth curves), a mixed-effects model of weight versus age was used, using genotype as fixed factors and subject (mice) as random factors, with analysis of variance (ANOVA) to test the overall effect of genotype on growth, followed by Tukey post-hoc tests. Standard significance levels were used: *$p < 0.05$, **$p < 0.01$, ***$p < 0.001$, ****$p < 0.0001$.

## Acknowledgements

We thank Jacques Drouin and Konstantin Khetchoumian (Institut de Recherches Cliniques de Montreal) and Barry Thompson (The Francis Crick Institute) for helpful discussions, and Lori Raetzman (University of Illinois) for sharing and discussing data. We are grateful to Dr. AF Parlow and the NHPP for reagents for the sandwich ELISA and immunofluorescence experiments. We thank Simon Rhodes (Indiana University School of Medicine) and Francis Poulat (Institut de Génétique Humaine, Montpellier) for the generous gift of antibodies. We also thank Biological Services and the light microscopy platform at the MRC National Institute for Medical Research and the Francis Crick Institute, for their assistance and technical support; and Pierre Fontanaud (Institut de Génomique Fonctionnelle, Montpellier) for valuable assistance with statistical analyses. This work was supported by the Medical Research Council, UK (U117512772, U117562207 and U117570590) and the Francis Crick Institute (grant 10107).

## Additional information

### Funding

| Funder | Grant reference number | Author |
| --- | --- | --- |
| Medical Research Council | U117562207 | Paul Le Tissier |
| Cancer Research UK | FC001107 | Robin Lovell-Badge |
| Medical Research Council | U117512772 | Robin Lovell-Badge |
| Medical Research Council | FC001107 | Robin Lovell-Badge |
| Wellcome | FC001107 | Robin Lovell-Badge |

The funders had no role in study design, data collection and interpretation, or the decision to submit the work for publication.

### Author contributions

Leonard Cheung, Data curation, Formal analysis, Investigation, Methodology, Writing—review and editing; Paul Le Tissier, Conceptualization, Supervision, Funding acquisition, Writing—review and editing; Sam GJ Goldsmith, Formal analysis, Performed Nkx3.1Cre lineage analysis; Mathias Treier, Resources, Generated the POU1F1-cre mice; Robin Lovell-Badge, Conceptualization, Supervision, Funding acquisition, Project administration, Writing—review and editing; Karine Rizzoti,

Conceptualization, Data curation, Formal analysis, Supervision, Investigation, Methodology, Writing—original draft, Writing—review and editing

### Author ORCIDs
Leonard Cheung http://orcid.org/0000-0002-0912-9594
Karine Rizzoti http://orcid.org/0000-0003-0711-5452

### Ethics
Animal experimentation: All experiments carried out on mice were approved under the UK Animal (scientific procedures) Act (Project licence 80/2405 and 70/8560).

### Decision letter and Author response
Decision letter https://doi.org/10.7554/eLife.33318.030
Author response https://doi.org/10.7554/eLife.33318.031

---

## Additional files

### Supplementary files
• Supplementary file 1. RT-qPCR primer sequences.
DOI: https://doi.org/10.7554/eLife.33318.027

• Transparent reporting form
DOI: https://doi.org/10.7554/eLife.33318.028

---

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
