## [Decision Letter]

Thank you for submitting your article "NOTCH activity differentially affects pituitary endocrine cell fate acquisition and maintenance" for consideration by *eLife*. Your article has been reviewed by three peer reviewers, and the evaluation has been overseen by a Reviewing Editor and Fiona Watt as the Senior Editor. The following individual involved in review of your submission has agreed to reveal his identity: Shannon Davis (Reviewer #1).

The reviewers have discussed the reviews with one another and the Reviewing Editor has drafted this decision to help you prepare a revised submission.

Summary:

This manuscript presents data that demonstrates that the NOTCH signaling pathway has a differential effect on cell lineages in the pituitary gland. Melanotropes in the pituitary intermediate lobe are sensitive to activation of NOTCH signaling, while anterior lobe cell types of the POU1F1 lineage are more resistant to NOTCH activation. It builds on previous work to highlight the timing of NOTCH signaling in regulating the transition from pituitary progenitors to determined cell lineages. This detailed study clearly situates the role of the NOTCH pathway in the life cycle of developing pituitary cells and is thus very important for the field. In particular, it corrects the false conclusions suggested by earlier work, particularly that of Zhu et al.

Essential revisions:

1) Since the Nkx3.1Cre line has not been used extensively in the pituitary gland please provide some lineage tracing data to show that the anterior lobe cell types are derived from Nkx3.1 expressing progenitors. This seems especially important when analysing the expression of lineage markers including Pou1f1 and hormones shown in Figure 1B. Do the gonadotrope progenitors express Nkx3.2. This is important for the interpretation of the Lhb expression levels.

2) Similarly, a novel Pou1f1Cre strain is used for the study but lacks validation via lineage tracing to confirm that Cre expression is faithful to the Pou1f1 lineage. This should be shown in the Supplementary data.

3) More data needs to be provided to bolster the argument that NICD expressing cells are extruded and incorporated into the anterior lobe. At present only a single example is provided. More quantification of the numbers of mutant versus wild type in which this phenomenon is seen is necessary.

4) Finally, to confirm the conclusion that E-cadherin and hence epithelial polarity is potentially disrupted, ideally further polarity markers should be used to confirm this finding in additional mutants as compared to wild type controls.

---

## [Author Response]

This manuscript presents data that demonstrates that the NOTCH signaling pathway has a differential effect on cell lineages in the pituitary gland. Melanotropes in the pituitary intermediate lobe are sensitive to activation of NOTCH signaling, while anterior lobe cell types of the POU1F1 lineage are more resistant to NOTCH activation. It builds on previous work to highlight the timing of NOTCH signaling in regulating the transition from pituitary progenitors to determined cell lineages. This detailed study clearly situates the role of the NOTCH pathway in the life cycle of developing pituitary cells and is thus very important for the field. In particular, it corrects the false conclusions suggested by earlier work, particularly that of Zhu et al.Essential revisions:1) Since the Nkx3.1Cre line has not been used extensively in the pituitary gland please provide some lineage tracing data to show that the anterior lobe cell types are derived from Nkx3.1 expressing progenitors. This seems especially important when analysing the expression of lineage markers including Pou1f1 and hormones shown in Figure 1B. Do the gonadotrope progenitors express Nkx3.2. This is important for the interpretation of the Lhb expression levels.

We have now included in Figure 1 a detailed lineage tracing analysis for the *Nkx3.1Cre* strain. We have performed lineage tracing in *Nkx3.1^Cre/+^; Rosa26^ReYFP/+^*pituitaries at 18.5dpc and quantified the percentage of eYFP;hormone positive cells for each endocrine lineage of the pituitary. This work was performed by Sam Goldsmith for his doctoral thesis in our lab, so we have now included his name in the author list and specified his contribution. This analysis shows that Nkx3.1Cre is mostly active in IL POMC positive melanotroph precursors, followed by POMC positive corticotrophs, and somatotrophs. These data support our results because we show that over-activation of NOTCH pathway using this Cre driver affects POMC expressing cells most significantly. Less than 40% of gonadotrophs are present in the progeny of Nkx3.1^Cre^ expressing cells and we did not observe a reduction in *Lhb* expression by RT-qPCR. However, we would expect that NOTCH activation in precursors blocks gonadotroph differentiation, and therefore a reduction in hormone levels. It is possible that such a defect affecting such a proportion of recombined cells is too low to detect an effect by RT-qPCR, and/or that cells of the lineage that did not recombine the NOTCH allele can compensate and restore normal numbers. In addition, determination of gonadotrophs starts at 12.5dpc as GnRHR is upregulated, which is the same stage as Nkx3.1^Cre^ is active. Therefore, it is alternatively possible that future gonadotrophs are already determined, and in consequence less sensitive to NOTCH activation, as we observed in the Pou1F1 lineage. Similarly, TSH levels are not affected but transient embryonic thyrotrophs that differentiate very early, at the time when Nkx3.1^Cre^ is active, would have differentiated before NICD is expressed.

In addition, this sentence is added in the Results section:

“Lineage tracing analysis of NKX3.1Cre confirms that this driver is predominantly active in future melanotrophs, and also, albeit less efficiently, in all anterior lobe endocrine populations (Figure 1A,B).”

2) Similarly, a novel Pou1f1Cre strain is used for the study but lacks validation via lineage tracing to confirm that Cre expression is faithful to the Pou1f1 lineage. This should be shown in the Supplementary data.

We have now performed lineage tracing for the *Pou1f1-Cre* strain at 18.5dpc. We have inserted a new figure, Figure 4—figure supplement 1. We show that there is some ectopic activity in the corticotroph and gonadotroph lineages, as we respectively observed some reporter;POMC and reporter;LH double positive cells. We analysed POU1F1-Cre progeny and observed that the vast majority of recombined cells are PIT1 positive 78.2%( ± 11.16SD, n=3) while we estimated the ectopic activity (reporter positive;PIT1 negative cells) represents 4.9% ( ± 2.5 SD, n=3).

3) More data needs to be provided to bolster the argument that NICD expressing cells are extruded and incorporated into the anterior lobe. At present only a single example is provided. More quantification of the numbers of mutant versus wild type in which this phenomenon is seen is necessary.

We apologize because the legend describing the quantification was really poor and gave the impression that this was not done properly. We have now amended the legend, added one more mutant embryo in the quantification and, importantly, added an annotated scheme to explain how the counting was done. We are confident this shows clearly that the quantification was performed adequately and provides a clear and significant account of the biased localization of recombined cells toward the pouch lumen.

Figure legend “In control *Sox2^CreERT2/+^;Rosa26^ReYFP^*pituitaries, eYFP positive cells are homogeneously distributed in IL (n=5 embryos, 58 ( ± 8.5SD) cells were counted/embryo). In contrast, 85% of NICD1iresGFP positive cells in mutant embryos are found in the ventral half of IL (n=5 embryos, 29 ( ± 23.6SD) cells were counted/embryo). Unpaired t test performed, p=0.0045 after angular transformation of percentages.”

4) Finally, to confirm the conclusion that E-cadherin and hence epithelial polarity is potentially disrupted, ideally further polarity markers should be used to confirm this finding in additional mutants as compared to wild type controls.

We have now examined the expression of ZO-1 localized at tight junctions in epithelial cells. We also observe a loss/aberrant localization of this protein, similarly to the alterations observed for Ecadherin, normally restricted to adherent junctions. These observations, made exclusively in NICD;GFP positive cells, reinforce our conclusions that epithelial polarity is lost in the NICD cells closest to Rathke’s pouch cleft. We are not excluding the possibility that this phenomenon happens in normal embryos, but even if it does, it is extremely rare as we hardly see these zones of apparent contact between the two epithelia. In contrast, these are frequent in our mutants and they correlate very well with the succeeding absence of mutant cells in IL. Because we are looking at mosaics, where most cells are effectively wild type and a much smaller proportion are mutant, we do not think it is necessary to look at purely wild type controls. Indeed, this type of mosaic analysis where only the mutant cells exhibit the phenotype is a robust way of addressing the question, eliminating variability between embryos. In addition, we have also examined expression of Slug, as NOTCH activation has been connected to EMT. We do not observe any up-regulation of Slug in NICD;GFP positive cells suggesting that EMT is not induced in these cells. Finally, we also examined expression of N-cadherin as there is a switch of expression from E-cadherin in epithelial cells, to N-cadherin in mesenchymal cells but N-cadherin was present throughout Rathke’s pouch and its pattern didn’t seem altered in NICD;GFP positive cells.

The results were modified as follows.

To assess epithelial integrity in mutant cells, we examined E-cadherin and ZO-1 expression, present respectively in adherent and tight junctions. Alterations in localisation of both proteins, normally present in the apical lateral membranes, are observed with a relocalisation around the plasma membrane, and sometimes loss from the apical ones, in IL NICD1iresnucGFP-positive cells closest to Rathke’s cleft. This suggests a loss of epithelial polarity of mutant cells (Figure 2E, F).